# Regulatory Processes in Populations of Forest Insects (A Case Study of Insect Species Damaging the Pine *Pinus sylvestris* L. in Forests of SIBERIA)

Vladislav Soukhovolsky [1], Tamara Ovchinnikova [1], Olga Tarasova [2], Yulia Ivanova [3] and Anton Kovalev [4,*]

[1] V. N. Sukachev Institute of Forest SB RAS, Akademgorodok 50/28, 660036 Krasnoyarsk, Russia
[2] Laboratory of Ecosystems Biogeochemistry, Institute of Ecology and Geography, Siberian Federal University, Av. Svobodny 79, 660041 Krasnoyarsk, Russia
[3] Institute of Biophysics SB RAS, Academgorodok 50/50, 660036 Krasnoyarsk, Russia
[4] Krasnoyarsk Scientific Center SB RAS, Akademgorodok 50, 660036 Krasnoyarsk, Russia
* Correspondence: sunhi.prime@gmail.com; Tel.: +7-9039236335

**Abstract:** The present study addresses the population dynamics of five species of phyllophagous forest insects in five habitats located in the Krasnoturansk pine forest (south Middle Siberia). Based on the data of insect surveys obtained during 1979–2016, autoregressive (AR) models of population dynamics have been proposed, with the current population density being dependent on population densities of the preceding years. Methods of calculation of the autoregression order and coefficients of AR equations have been presented. The study shows that, for different insect species in different habitats, the lags between the current population density and the densities of the previous years are not the same. AR equations characterize positive and negative feedbacks regulating population dynamics. By using AR equations, up to 90% of population density variance can be taken into account. Stability margin, which is calculated from coefficients of AR models, has been proposed as a parameter to assess the stability of population dynamics. A small stability margin indicates a high risk of outbreak of an insect species.

**Keywords:** insect; population dynamics; outbreaks; autoregulation; positive and negative feedbacks; modeling; stability margin





## 1. Introduction

It is important to study relationships in the population dynamics of different forest insect species in the same habitat or the same species in different habitats, in order to be able to make an accurate estimate of the effects of various factors on population dynamics. Such relationships indicate the presence of ecological mechanisms leading to the coordination of time series in population dynamics, enabling the assessment of various factors affecting these populations.

In recent decades, spatiotemporal synchrony in the population dynamics of different species of forest insects has been studied by many researchers. Insect population dynamics in habitats with similar climatic but different landscape conditions has repeatedly served as an object of analysis. The analyses have shown that there is significant progress in quantifying temporal population dynamics and understanding the underlying mechanisms [1,2]. Much less is known about the causes of complex spatial changes that often characterize outbreaks. According to the epicenter hypothesis, an insect outbreak begins in certain geographical foci and spreads to adjacent areas. This hypothesis has been advanced for many forest insect species [3–6]. Landscape mosaics and spatial alternations infavorable and unfavorable habitats can also influence the development of foci [7,8]. Contrary to the epicenter hypothesis, the analysis of field data suggested that the spread of a focus during outbreaks arises from few epicenters in areas with a high insect concentration, and favorable habitat and landscape features [9].

When analyzing spatial synchrony and the correlation between population dynamics in local or large geographic ranges, it was shown that Moran's effect as a whole seemed to be the dominant process regarding the spatial dynamics of these species. However, the analysis of gypsy moth outbreak data showed that local dynamics significantly varied, which violates the assumptions of the Moran model about the identity of spatially distributed populations' dynamic properties [10].

When studying the spatial synchrony of 10 Lepidoptera species depending on three weather variables (minimum temperature, maximum temperature, precipitation) at 12 sites, measured using cross-correlation functions, it was shown that local weather conditions have a synchronizing effect on the phyllophagous insects' population dynamics [11].

The strength of the relationship between population dynamics of the same species in different habitats monotonically decreases as the distance between these habitats increases. If the strength of relationships in population dynamics does not decrease with increasing distance between habitats, and the distance between them is considerably longer than the movement radius of the individuals of the species, this should be regarded as the spatial coherence of the species, associated with the population response to a powerful modifying factor [12]. Numerous quantitative data suggest direct effects of the weather and interannual climate variability, which induce large-scale outbreaks [13–16]. Over longer periods of time (from decades to centuries), climate change may indirectly affect insect abundance [17].

Recent studies demonstrate that outbreaks of forest defoliators are more complex events than previously thought. For instance, outbreaks are caused by factors at different levels, i.e., host numbers at the tree stand level, correlations at the landscape level, and climate effects at the regional level [18,19]. Hence, there is a need for better quantitative insights into cause-and-effect relationships and inter-scale correlations that lead to large-scale outbreaks [18,20]. Various authors are of the opinion that synchrony in population dynamics of the same species in different habitats may be caused by the Moran effect, which is associated with the climate similarity over a vast area and similar responses of the populations to weather change in different habitats [21–28].

Quantification of the relationships in population dynamics of the same species in different habitats or several species in the same habitat should be based on the long-term (decades-long) monitoring of these insect species on permanent sample plots. Unfortunately, such studies are not conducted frequently enough, and no accurate statistical estimates of their correlations can be obtained for short time series [29–32].

However, an analysis of the time series of insect population dynamics can be replaced by an analysis of the regulatory characteristics of the strength and characteristic times of the regulation feedbacks for individual populations. In the present study, the regulatory characteristics of forest insect population dynamics are estimated using the data from surveys of phyllophagous insects in pine tree stands in south Krasnoyarsk Krai (Russia) conducted from 1979 to 2016.

The populations of forest insects in certain habitats were considered to be temporally stationary systems with a set of positive and negative feedbacks, which depended on various factors and constituted the population dynamics. The conceptual scheme of this approach is shown in Figure 1.

Feedback loops reflect the influence of an object's past states on its current state. For the studied systems, it was assumed that the feedback loops in question are linear and that a feedback signal can be expressed as $z(i) = a \cdot y(i - \tau)$, with delay $\tau$ and feedback amplitude $a > 0$ for positive feedback and $a < 0$ for negative feedback. The number of feedback loops of different signs (positive or negative) was unknown, but in the general case, considering the feedback loops, a steady-state system can be described as autoregressive (AR). This can be written via the following equation:

$$y(i) = \sum_{j=1}^{k} a_j y(i - j) + \varepsilon \tag{1}$$

According to (1), in order to describe regulatory feedbacks in the system, it is necessary to know the total number of significant feedbacks, $k$, and absolute values and signs of feedback coefficients $a_1, \ldots a_k$. The present study is devoted to analysis of the feedback systems in regulation of population dynamics of different species in different habitats.

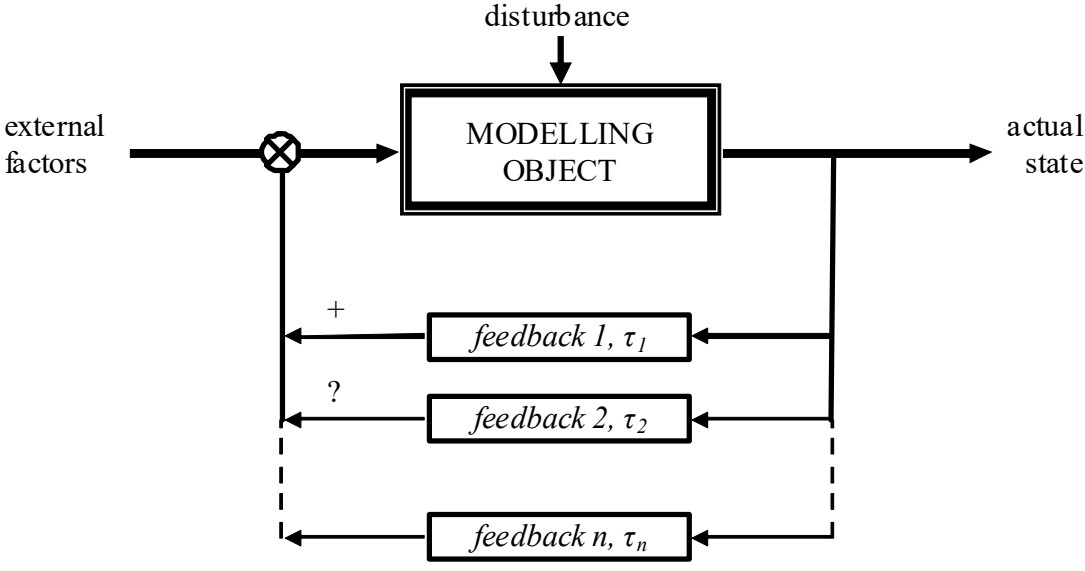

**Figure 1.** Conceptual scheme of population model ($\tau 1, \ldots, \tau n$–characteristic feedback times).

## 2. Materials and Methods

Population dynamics of phyllophagous insects in various landscape structures of the Krasnoturansk ribbon pine forest (south Middle Siberia, 54°16.315′N, 91°37.757′E) were studied for almost 40 years, from 1979 to 2016 [33,34]. The main forest-forming tree species of the ribbon forest is Scots pine *Pinus sylvestris* L. When the studies were started, the trees in the pine stand were 60–80 years old. Insect habitats in the Krasnoturansk pine forest were classified according to landscape characteristics. Different insect habitats differ in microrelief as well as seasonal temperature and humidity regimes. Further in the work, following conditional names will be given: 1) peaks and straight slopes of the hills–"Top"; 2) narrow flatlands adjacent to the tops of the hills–"Plakor"; 3) slopes of the hills; 4) steep slopes of valleys and ravines; 5) slight slopes of southern and western exposure–"Dune"; 6) gentle inclined terrace-like surfaces–"Terrace"; 7)flat proluvial wet lowlands–"Lake". Figure 2 shows the spatial structure of habitats in the Krasnoturansk pine forest.

Table 1 lists distances between sample plots in different habitats.

**Table 1.** Distances (km) between sample plots in different areas of Krasnoturansk pine forest.

| Habitat | Habitat | | | | |
|---|---|---|---|---|---|
|  | **Plakor** | **Top** | **Dune** | **Terrace** | **Lake** |
| Plakor | 0.00 | 1.6 | 1.2 | 1.8 | 2.9 |
| Top |  | 0.00 | 2.3 | 2.1 | 3.8 |
| Dune |  |  | 0.00 | 1.0 | 3.9 |
| Terrace |  |  |  | 0.00 | 4.7 |

Phyllophagous insects inhabiting the Krasnoturansk pine forest are typical for pine forests of Siberia: *Bupaluspiniarius* L., *Semiothisaliturata* Cl. (Geometridae), *Dendrolimuspini* L. (Lasiocampidae), and two sawflies–*Gilpinia virens* Kl. and *Microdiprionpallipes* Fall. (Diprionidae). During 1979–2016, population densities of those species were determined in the first ten days of August by beating the trees to knock off the insects into a sheet

spread on the ground. Annual surveys of defoliating insect populations were conducted in five types of habitats: "Top", "Plakor", "Dune", "Terrace", and "Lake". The data on population densities of the study species in different habitats were reported elsewhere [34]. Table 2 gives the long-term annual average densities (insects per tree) of the study species populations, standard deviations of the mean, and maximal densities in different habitats.

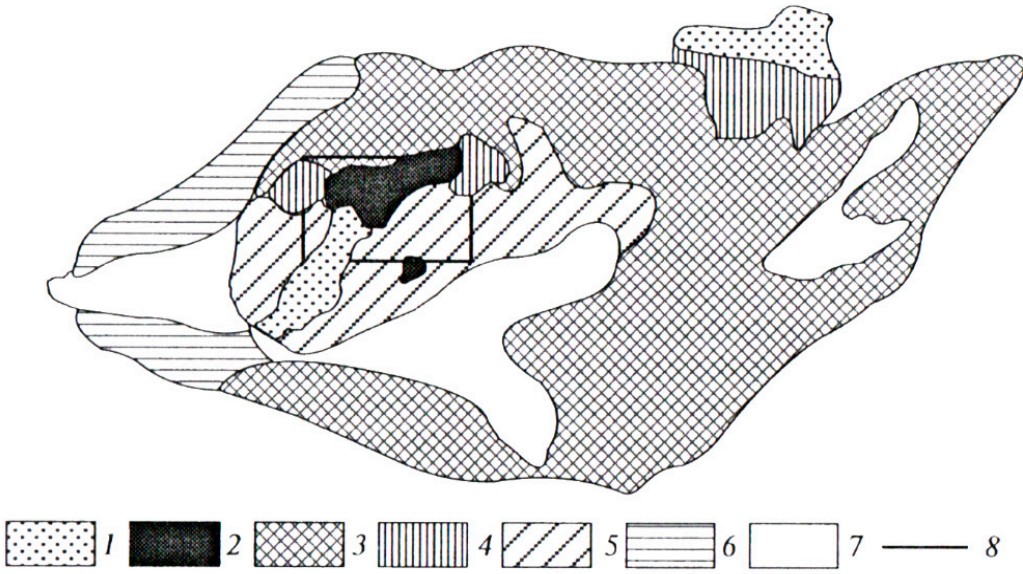

**Figure 2.** Landscape structure of the Krasnoturansk pine forest [33]. (1) Habitat "Top"; (2) Habitat "Plakor"; (3) Slopes of the hills; (4) Steep slopes of valleys and ravines; (5) Habitat "Dune"; (6) Habitat "Terrace"; (7) Habitat "Lake"; (8) Boundaries of the *B. piniarius* outbreak in 1973–1977.

Total population densities of the defoliating species in all habitats were low during the entire study period: no more than 13.51 insects per tree (Table 2). Population densities of phyllophagous insects were not always as low as that, and even before the surveys were started, in 1976–1978, a pine looper outbreak occurred in the Krasnoturansk pine forest, and a considerable number of Scots pine (*P. sylvestris*) trees were damaged [33].

**Table 2.** Long-term annual average densities (insects per tree) of the study species populations, standard deviations of the mean, and maximal densities in different habitats.

| Habitat | Parameters | Species | | | | |
|---------|-----------|---------|---------|---------|---------|---------|
| | | *B. piniarius* | *S. liturata* | *G. virens* | *M. pallipes* | *D. pini* |
| Top | Average density <N> | 0.72 | 0.09 | 0.14 | 0.10 | 0.16 |
| | standart deviations | 1.96 | 0.14 | 0.22 | 0.15 | 0.28 |
| | maximal density Nmax | 10.88 | 0.55 | 1.00 | 0.60 | 1.30 |
| Plakor | Average density <N> | 0.68 | 0.05 | 0.13 | 0.06 | 0.15 |
| | standart deviations | 1.28 | 0.07 | 0.23 | 0.11 | 0.27 |
| | maximal density Nmax | 5.60 | 0.24 | 1.10 | 0.45 | 1.20 |
| Lake | Average density <N> | 0.18 | 0.19 | 0.30 | 0.15 | 0.09 |
| | standart deviations | 0.31 | 0.38 | 0.59 | 0.41 | 0.13 |
| | maximal density Nmax | 1.56 | 2.05 | 2.68 | 2.25 | 0.65 |
| Terrace | Average density <N> | 0.11 | 0.17 | 0.14 | 0.04 | 0.38 |
| | standart deviations | 0.18 | 0.30 | 0.31 | 0.06 | 0.77 |
| | maximal density Nmax | 0.76 | 1.56 | 1.36 | 0.24 | 3.08 |

**Table 2.** *Cont.*

| Habitat | Parameters | Species | | | | |
| --- | --- | --- | --- | --- | --- | --- |
| | | *B. piniarius* | *S. liturata* | *G. virens* | *M. pallipes* | *D. pini* |
| Dune | Average density <N> | 1.46 | 0.16 | 0.20 | 0.05 | 0.33 |
| | standart deviations | 3.39 | 0.25 | 0.25 | 0.08 | 0.64 |
| | maximal density Nmax | 13.51 | 1.32 | 0.92 | 0.40 | 2.96 |

## 3. Results

The result of this study is the procedure of calculating regulatory characteristics of phyllophagous insect population dynamics, which can be used to investigate the population dynamics of these insects and assess the risks of outbreaks of individual species.

According to (1), regulatory characteristics of population dynamics can be estimated from parameters of autoregressive (AR) equations calculated based on the known population densities measured during the surveys. If the density values of population dynamics are known and population dynamics is characterized by steady state, at a known magnitude of the number of significant feedbacks, $k$, Equation (1) can be considered a linear regression equation relative to the unknown parameters $a_1, \ldots a_k$. Then, these coefficients can be determined using conventional methods of linear regression analysis. Thus, the AR model of population density of an individual species was constructed using the following procedure:

1. The studied time series was tested for stationarity and, if necessary, transformed to the stationary time series.
2. The order of the AR equation, characterizing the number of significant feedbacks, was estimated.
3. Based on the results of steps 1 and 2, absolute values and signs of feedback coefficients were calculated, using the data of long-term surveys.
4. Reliability of the calculations was assessed.

This method includes:

- "Repair" of the time series of population dynamics and replacement of the zero values of population densities with values that are smaller by a factor of two than the minimal values of the measured insect population densities for the entire measurement period;
- Conversion to logarithmic scale in order to reduce data scattering (the replacement of the zero density values in the step above forms the prerequisite);
- Filtering the high-frequency component of the time series in order to reduce the errors in insect surveys. The Hann window is used [35]:

$$y(i) = 0.24 \cdot ln\ x(i-1) + 0.52 \cdot ln\ x(i) + 0.24 \cdot ln\ x(i+1) \qquad (2)$$

- The partial autocorrelation function, PACF, is calculated [36] to determine the order k of autoregressive Equation (1). The order of autoregressive equation is characterized by the maximal PACF;
- Model (2) is considered as a regression equation with {y(i)} values known from survey data, and conventional methods are used to determine the unknown coefficients a0, . . . ., ak;
- The accuracy of the calculations using the model equation of insect population dynamics is estimated from the value of coefficient of determination, $R^2$, characterizing the portion of variance of logarithmic insect abundance calculated using the AR model; the significance of coefficients of the AR model is estimated by Student's *t*-test and Fisher's test [26];
- Assessment of the synchrony of the time series of transformed data and the model time series is based on the value of the cross-correlation function (CCF) [30]. For synchronous time series, the maximal value of CCF ($k = 0$) is close to 1;

- The oscillation of the time series of transformed data are estimated from the time series range [31,32,36];
- The stability of the time series of population dynamics for a given habitat is estimated from the stability margin η calculated using coefficients $a_1, \ldots, a_k$ of model (2) for this habitat. The greater the stability margin η, the higher the stability of the dynamics series. The calculation procedure is given in Supplement S3.

Calculations were performed for all time series of all insect populations in all habitats. The data of insect surveys are presented in Supplement S1.

Analysis of the time series of population dynamics of the study forest insects showed that all the time series could be regarded as stationary. Figure 3 shows a typical curve in *B. piniarius* population dynamics in Habitat Terrace (Curve 1).

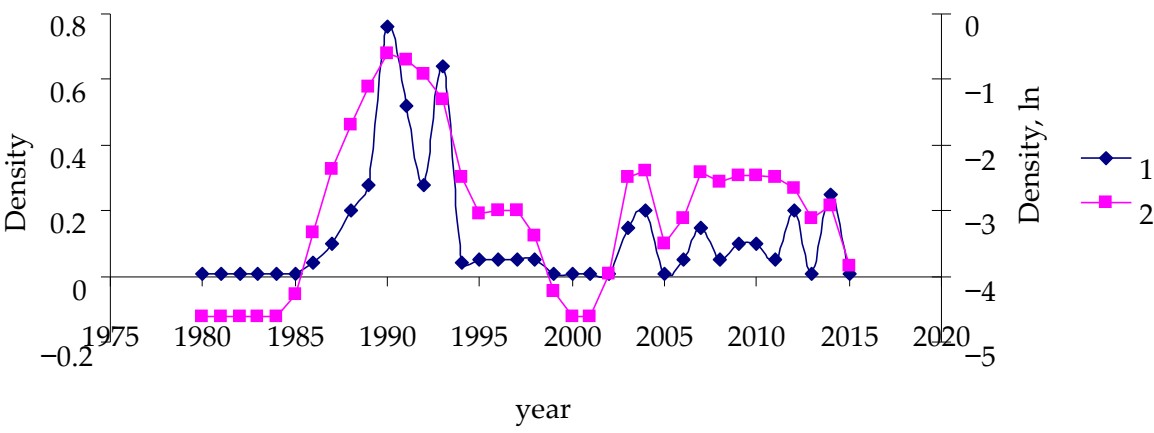

**Figure 3.** The time series of the field data (1) and the series of the transformed data (2) of *B. piniarius* population dynamics in Habitat Terrace.

After "repair", conversion to the log scale, and filtering, the time series in Figure 3 is transformed to Curve 2 in Figure 3. All time series in this study can be characterized as temporally stationary. The stationarity of the time series of population dynamics is a prerequisite for considering AR models as a tool for characterizing population dynamics. In order to determine the order of autoregression, partial autocorrelation functions of transformed series were calculated.

Figure 4 shows partial autocorrelation function of Curve 3 in Figure 3.

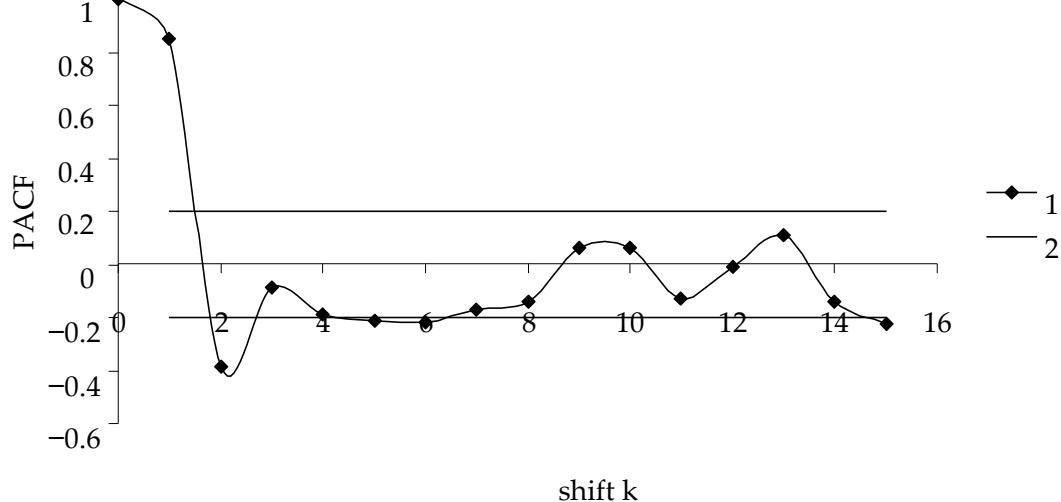

**Figure 4.** Partial autocorrelation function of the transformed time series of *B. piniarius* population dynamics in Habitat Terrace.

The order of autoregression is 2 (Figure 4), i.e., regulation of the time series is characterized by the presence of two feedbacks with characteristic times of 1 and 2 years. For the specified value $k = 2$, coefficients of regression equation were calculated from the values of the transformed series. Table 3 presents the values of coefficients and the significance of these values for *B. piniarius* in Habitat Terrace.

**Table 3.** Coefficients of the AR Equation (2), stability margin η and significance of these coefficients for *B. piniarius* in Habitat Terrace.

| Variables | Coefficient | Std.Err. | t(29) | *p*-Value |
|:---:|:---:|:---:|:---:|:---:|
| a0 | −0.565 | 0.235 | −2.406 | 0.022 |
| y(i-2) | −0.530 | 0.155 | −3.427 | 0.002 |
| y(i-1) | 1.339 | 0.159 | 8.410 | 0.000 |
| adjR$^2$ | 0.820 | | | 0.000 |
| F | 74.400 | | | |
| η | 0.08 | | | |

The AR model is characterized by positive feedback between y(i) and y(i − 1) (coefficient a(i − 1) > 0) and negative feedback between y(i) and y(i − 2) (coefficient a(i − 2) <0) (Table 3). All coefficients of the AR model are significant (Table 3). Coefficient of determination R$^2$ characterizes the contribution of feedback systems to the variance of the values of transformed population density. The value of $(1 − R^2)$ characterizes the contribution of other factors (such as weather) to insect population dynamics. Figure 5 demonstrates correspondence between transformed and model time series of *B. piniarius* population dynamics in Habitat Terrace.

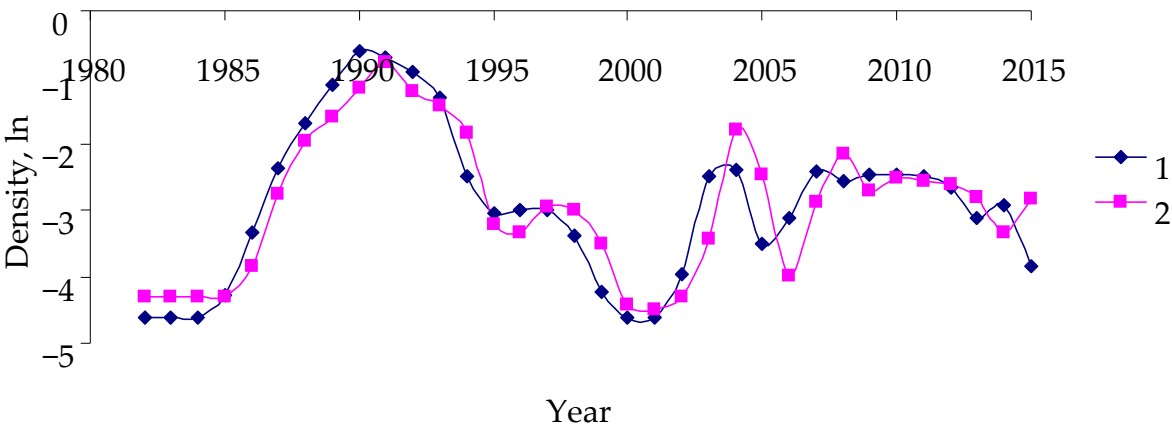

**Figure 5.** Transformed (1) and model (2) time series of *B. piniarius* population dynamics in Habitat Terrace.

Synchrony of the transformed and model series of *B. piniarius* population dynamics was estimated by calculating cross-correlation function (CCF) between these time series (Figure 6).

The maximal CCF is achieved at a shift of $k = 0$, indicating synchrony of these time series (Figure 6). The same procedure was conducted for all 25 studied time series (five time series for each of the five habitats). The calculation details are given in Tables S1K2–S5K2 of Supplement S2.

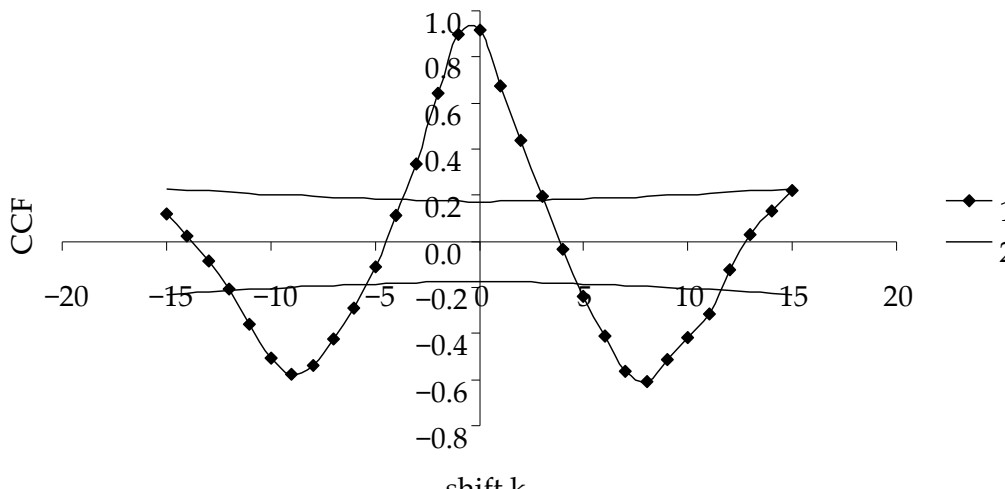

**Figure 6.** Cross-correlation function of the time series of transformed field data and model data for *B. piniarius* in Habitat Terrace.

## 4. Discussion

For three species (*B. piniarius*, *S. liturata*, and *D. pini*), population dynamics is characterized by the presence of one statistically significant positive feedback and one statistically significant negative feedback (Tables S1K2–S5K2, Supplement S2). The regulatory relationships remain the same at a distance of 4 km between sample plots (Table 1). Positive feedback can be described as the influence of the parent generation on the current density, and negative feedback, with a two-year lag, can reflect the effect of parasites on the population. For *G. virens* in four of the five habitats, autoregression is characterized by the presence of one positive feedback (with a one-year lag) and one negative feedback (with a two-year lag)–the same as for *B. piniarius*, *S. liturata*, and *D. pini*. However, for one habitat of *G. virens* (Top), the AR model is characterized by the presence of four feedbacks with from one- to four-year lags. For *M. pallipes*, the situation is the most complex: in Habitat Dune, the AR model is characterized by the presence of two feedbacks (positive and negative), in two habitats (Plakor and Terrace), the AR model includes two positive and one negative feedbacks), and, finally, in Habitats Top and Lake, the AR model includes two negative and two positive feedbacks.

In models for *B. piniarius*, the contribution of autoregressive components to the variance of the data on population density varies between 82 and 94%, and in models for *S. liturata*, the contribution ranges from 80 to 90%. The contributions of the AR model to population dynamics of *D. pini* are almost the same in different habitats. For *M. pallipes*, the contribution of AR components to population dynamics ranges between 75 and 82%. The smallest contribution of autoregressive components (61.5 to 71.5%) is observed in the model for *G. virens*.

In terms of control theory, the maintenance of any system (including the insect system in the forest) in stable state is related to the effects of feedbacks [37]. The strong and fast negative feedbacks enable the system subjected to impact to return to its normal state quickly. By contrast, strong and fast positive feedbacks result in considerable disturbance to the system. If both negative and positive feedbacks are present, as is typical for insects in the Krasnoturansk pine forest, the situation becomes unclear, and the assessment of the stability of the system should be based on integrated parameters, which would consider the effects of both negative and positive feedback. In the control theory, stability margin, η, is used as such an integrated parameter [38]. The stability margin η is calculated from the values of coefficients of the AR model, which characterize both positive and negative feedbacks. The procedure of calculating the stability margin is given in Supplement S3. The stability margins η for all insect species in every habitat are listed in Table 4. Table 4

provides average stability margins for all habitats for individual species and average stability margins for all species in individual habitats.

**Table 4.** Average stability margins for all habitats for individual species <ηs> and average stability margins for all species in individual habitats <ηm>.

| Species | Average Stability Margin η for a Species | Habitat | Average Stability Stability Margin η for a Habitat |
|---------|------------------------------------------|---------|----------------------------------------------------|
| *B. piniaius* | 0.13 | Top | 0.38 |
| *S. liturata* | 0.12 | Plakor | 0.21 |
| *G. virens* | 0.40 | Lake | 0.20 |
| *M. pallipes* | 0.46 | Terrace | 0.28 |
| *D. pini* | 0.12 | Dune | 0.16 |

*G. virens* and *M. pallipes* were found to be the most stable species (Table 4). For three other species–*B. piniarius*, *S. liturata*, and *D. pini*–the stability margin is smaller by a factor of two-three. This is indicative of the high risk of density variations in these species and the probability of their outbreaks. Analysis of the average stability margins for species in individual habitats shows that Dune is the least stable habitat while Top is the most stable.

The larger the stability margin of the system, the less likely any outliers–population density increases or decreases–are to occur during the development of the population (Figure 7).

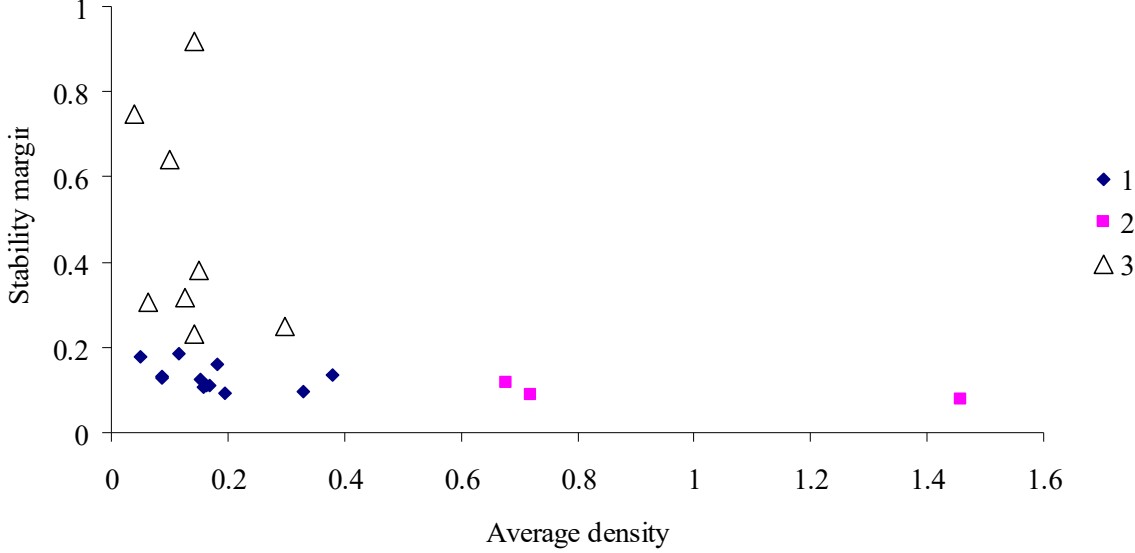

**Figure 7.** The relationship between long-term annual average population density and stability margin (1–*S. liturata*, *D. pini*; *B. piniarius* in HabitatsLake and Terrace; 2–*B. piniarius* in habitats Top, Plakor, and Dune; 3–*G. virens*, *M. pallipera*).

For *B. piniaius*, an outbreak species in Habitats Top, Plakor, and Dune in the Krasnoturansk pine forest, with a stable sparse long-term annual average population density (over 0.6 larva per tree), the stability margin is low (η< 0.12). By contrast, for *G. virens* and *M. pallipera*, which are characterized by mere increases in density, at low long-term annual average population densities, the stability margin is rather large (η> 0.2). For intermediate species (*S. liturata* and *D. pini*) and *B. piniarius* in HabitatsLake and Terrace, where no outbreaks occur, at low population densities, the stability margins are small as well.

Thus, even in the period between outbreaks, when population density is relatively low, stability margins can be used to identify in plane {<N>, η} three clusters differing

in population dynamics and regulatory properties of the populations, which determine stability margin, and reveal outbreak prone species and habitats.

## 5. Conclusions

The proposed procedure was used to determine the coefficients of autoregressive models of population dynamics for all phyllophagous insect species in all habitats studied in this work. For most populations in different habitats, intra-population regulatory relationships are characterized by the presence of one positive feedback with a one-year lag and one negative feedback with a two-year lag. However, for *G. virens* and *M. pallipes*, the characteristic time of feedback lag reaches four years. When both positive and negative feedbacks are present, it is difficult to assess the stability of population dynamics. Thus, the stability margin is used to quantitatively assess the probability of the system falling out of the stability range. Calculations of stability margins for different insect species in different habitats show that, for *G. virens* and *M. pallipes*, stability is higher than for the other species studied in this work. *B. piniarius* populations in the habitats where outbreaks of this species occurred before the surveys have the smallest stability margin.

**Supplementary Materials:** The following supporting information can be downloaded at: https://www.mdpi.com/article/10.3390/d14121038/s1, Supplement S1—Insect's population dynamic, Supplement S2—Calculations of autoregressive functions for 25 models of insect population dynamics in five habitats in the Krasnoturansk pine forest, Supplement S3—stability margin calculation method [39].

**Author Contributions:** Conceptualization, V.S. and O.T.; methodology—V.S., T.O. and A.K.; software—A.K. and T.O.; data curation—V.S., O.T. and Y.I.; writing—original draft preparation—V.S.; writing—review and editing, V.S., Y.I., visualization—A.K. All authors have read and agreed to the published version of the manuscript.

**Funding:** This research was funded by Russian Scientific Foundation (research project number 22-24-00148).

**Institutional Review Board Statement:** Not applicable.

**Informed Consent Statement:** Not applicable.

**Conflicts of Interest:** The authors declare no conflict of interest.

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
