# Peer review of "Regulatory Processes in Populations of Forest Insects (A Case Study of Insect Species Damaging the Pine Pinus sylvestris L. in Forests of SIBERIA)"

_diversity, doi:10.3390/d14121038_

Round 1

Reviewer 1 Report

This is an interesting paper on a relevant topic. Here are my suggested edits:

- All botanical names of plants and insects should be italicized in the text and in tables or figure captions.

- Figure 2 needs additional labeling; it's not clear what is being shown by the different shapes with lines/cross-hatching.

- Are there relevant ecological differences between the different habitats despite being dominated by the same tree species?

- Table 2 formatting could be improved with lines separating rows of the different habitats, and it is unclear why Top is in bold with a line when the other habitats are not.

- Lines 115-119 would be better suited for the Results section instead of Materials and Methods. Furthermore, much of what is included in the Results sections is more appropriate for Materials and Methods and elaboration on the modelling outcomes should be included in the results.

- I do not see the stability margins for all insects species in every habitat in Table 3 (as is stated in line 254); they do appear to be in Supplement 2.

Author Response

Dear Reviewer, We tried to take into account all your comments:

 - All botanical names of plants and insects should be italicized in the text and in tables or figure captions.

Corrected

- Figure 2 needs additional labeling; it's not clear what is being shown by the different shapes with lines/cross-hatching.

Corrected and added description of various habitats in the text above.

- Are there relevant ecological differences between the different habitats despite being dominated by the same tree species?

Corrected in description of habitats

- Table 2 formatting could be improved with lines separating rows of the different habitats, and it is unclear why Top is in bold with a line when the other habitats are not.

Corrected

- Lines 115-119 would be better suited for the Results section instead of Materials and Methods. Furthermore, much of what is included in the Results sections is more appropriate for Materials and Methods and elaboration on the modelling outcomes should be included in the results.

Corrected. Sections Results and Materials and Methods restructured

- I do not see the stability margins for all insects species in every habitat in Table 3 (as is stated in line 254); they do appear to be in Supplement 2.

Corrected. Header and variable names in table 3 changed

Thank you for your attention. See revised version of the manuscript.

Reviewer 2 Report

The authors present a work using autoregressive models to describe long-term population data of forest insects. While the work is interesting, the organization of the manuscript is distracting and detracts from the overall work. Primarily, the authors must do more to establish the use of these AR models and the impact that the results can have on understanding this particular dataset. Also see below for more specific comments.

Lines 36-37: Please elaborate with 1-2 sentences of examples of this. Just adding 22 citations with no concrete examples doesn't give much for the reader to grasp.

Lines 38-46: I feel like you go from discussing distance based metrics and population dynamics to outbreaks with no clear connection between the two. And then on to climate change without solid transitions.

Lines 115-119: This seems more like results than methods.

Results: All these results seem more like methods. It's a step-by-step of how you did the work. I don't see any results here.

Line 175: Italicize species names. This is a problem throughout the manuscript.

Discussion until line 220: This is all results. Please reformat.

Author Response

Dear Reviewer, We tried to take into account all your comments:

We have significantly changed literature review. Ratio of results and discussion has also been changed. Italicize species names - corrected.

Regarding the following remark:

Lines 115-119: This seems more like results than methods.

All field observations were taken from our other publication [34] and represent source material for this paper.

Thank you for your attention. See revised version of the manuscript.

Round 2

Reviewer 2 Report

No further comments at this time.